# Machine Learning Approaches to Differentiate Sellar-Suprasellar Cystic Lesions on Magnetic Resonance Imaging

**DOI:** 10.3390/bioengineering10111295

**Published:** 2023-11-08

**Authors:** Chendan Jiang, Wentai Zhang, He Wang, Yixi Jiao, Yi Fang, Feng Feng, Ming Feng, Renzhi Wang

**Affiliations:** 1Department of Neurosurgery, Chinese Academy of Medical Sciences and Peking Union Medical College, Peking Union Medical College Hospital, Beijing 100730, China; 201301014@student.pumc.edu.cn (C.J.); zhangwt@student.pumc.edu.cn (W.Z.); wanghe15@student.pumc.edu.cm (H.W.); jiaoyx19@student.pumc.edu.cn (Y.J.); farry92@163.com (Y.F.); wangrenzhi@pumch.cn (R.W.); 2Department of Neurosurgery, Xuanwu Hospital, Capital Medical University, China International Neuroscience Institute, Beijing 100053, China; 3Department of Thoracic Surgery, Peking University First Hospital, Beijing 100034, China; 4Department of Radiology, Chinese Academy of Medical Sciences and Peking Union Medical College, Peking Union Medical College Hospital, Beijing 100730, China

**Keywords:** machine learning, radiomics, magnetic resonance imaging, cystic lesions

## Abstract

Cystic lesions are common lesions of the sellar region with various pathological types, including pituitary apoplexy, Rathke’s cleft cyst, cystic craniopharyngioma, etc. Suggested surgical approaches are not unique when dealing with different cystic lesions. However, cystic lesions with different pathological types were hard to differentiate on MRI with the naked eye by doctors. This study aimed to distinguish different pathological types of cystic lesions in the sellar region using preoperative magnetic resonance imaging (MRI). Radiomics and deep learning approaches were used to extract features from gadolinium-enhanced MRIs of 399 patients enrolled at Peking Union Medical College Hospital over the past 15 years. Paired imaging differentiations were performed on four subtypes, including pituitary apoplexy, cystic pituitary adenoma (cysticA), Rathke’s cleft cyst, and cystic craniopharyngioma. Results showed that the model achieved an average AUC value of 0.7685. The model based on a support vector machine could distinguish cystic craniopharyngioma from Rathke’s cleft cyst with the highest AUC value of 0.8584. However, distinguishing cystic apoplexy from pituitary apoplexy was difficult and almost unclassifiable with any algorithms on any feature set, with the AUC value being only 0.6641. Finally, the proposed methods achieved an average Accuracy of 0.7532, which outperformed the traditional clinical knowledge-based method by about 8%. Therefore, in this study, we first fill the gap in the existing literature and provide a non-invasive method for accurately differentiating between these lesions, which could improve preoperative diagnosis accuracy and help to make surgery plans in clinical work.

## 1. Introduction

Cystic lesions of the sellar-suprasellar region are a group of pathological changes in the pituitary glands or their adjacent structures. They could be primary lesions, including Rathke’s cleft cysts, cystic craniopharyngiomas, abscesses, arachnoid cysts, etc., or secondary changes from substantial lesions, including cystic pituitary adenomas (cysticA) and pituitary apoplexy [1]. In a random autopsy study involving 1000 cases, 113 cases of Rathke’s cleft cyst were found, with 37 cases with a diameter of at least 2 mm, which is higher than pituitary adenomas (31 cases) [2].

Although Transsphenoidal surgery is the primary surgical approach for treating sellar lesions, different surgical strategies are required for various sellar lesions. In addition, the optimistic therapeutic pathways remain controversial [3,4] and the pathological type is the basis of clinical decision-making. For cystic pituitary adenomas, total resection is the primary surgical approach. However, for Rathke’s cleft cysts and arachnoid cysts, it is recommended to perform fenestration and partial resection. In the case of cystic craniopharyngiomas, surgery should involve the removal of the cystic wall structure to prevent recurrence [4,5,6].

The pituitary gland, as an endocrine organ in the human body, can be differentially diagnosed for sellar lesions through blood hormone screening. However, in the case of cystic lesions in the sellar region, they typically present as cystic masses without causing abnormal hormone secretion levels, which increases the difficulty of making a differential diagnosis. As shown in Figure 1, different cystic lesions in the sellar region performed similarly in the MR image. Nonetheless, this requires clinical physicians to have sufficient experience, and currently, achieving an accurate preoperative diagnosis remains challenging.

Current research on sellar cystic lesions focuses more on the qualitative description of MRI, which is also the most widely used method in clinical practice. Different components (such as differences in protein percentage) in lesions show different signal strengths [7]. These known imaging features are the basis for the feasibility of automatic identification through quantitative analysis.

Radiomics, a recently emerging technique, extracts high-throughput features from digital imaging to quantitatively describe lesion characteristics, which builds a bridge between medical imaging and tumor phenotypes [8,9]. With the help of machine learning algorithms and artificial neural networks, investigating the intrinsic characteristics of high-throughput image features is easier and more systematic. Previous studies focused on malignant central nervous tumors with radiomic approaches, which showed charming performances on molecular subtype distinguishment [10,11]. Previous studies have investigated the use of radiomics in pituitary diseases, while few studies have focused specifically on cystic sellar lesions [12,13].

The present study aims to differentiate between the four most common types of cystic sellar lesions, namely Rathke’s cleft cysts, cystic craniopharyngiomas, cystic pituitary adenomas, and pituitary apoplexy, using radiomics and machine learning. The main contributions are as follows:A large number of patients diagnosed with cystic sellar lesions were enrolled to fill the gap in the existing literature and provide a non-invasive method for accurately differentiating between these lesions.Paired imaging differentiations were performed on four subtypes, and the model achieved an average AUC value of 0.7685.The model achieved an average accuracy of 0.7532, which outperformed the traditional clinical knowledge-based model by approximately 8%.

## 2. Related Works

### 2.1. Basic Characteristics of Cystic Sellar Lesions

Cystic Pituitary Adenomas or Pituitary Apoplexy: Approximately 48% of pituitary adenomas are cystic, which can be categorized as hemorrhagic or ischemic cystic pituitary adenomas, with hemorrhagic ones termed pituitary apoplexy. Cystic pituitary adenomas often exhibit thickened cyst walls with accompanying fluid content. Internal septations are commonly observed within the cyst, and the cystic changes tend to be eccentric within the pituitary [14].

Rathke’s Cleft Cysts: Rathke’s cleft cysts originate from Rathke’s cleft and are found in approximately 11% to 33% of patients undergoing pituitary biopsy. These cysts are typically smaller than 2 mm and are located in the midportion of the pituitary. They appear as round, thin-walled enhancements without internal septations or calcifications [14].

Craniopharyngioma: Due to the various origins of craniopharyngiomas, the resulting cysts can be located within or above the Sella turcica and may exhibit calcifications and multilocularity [15].

### 2.2. Clinical Knowledge-Based Method

Based on clinical prior knowledge, researchers have proposed a simple diagnostic approach [16]. Firstly, categorization is based on the characteristics of the cyst: cysts with calcifications are classified as craniopharyngiomas, those with fluid content as cystic pituitary adenomas, and those with intracystic nodules as Rathke’s cleft cysts. Subsequently, the assessment considers enhancement characteristics: absence of enhancement indicates arachnoid cysts, thin and regular enhancement suggests Rathke’s cleft cysts, thick-walled enhancement with septations suggests cystic pituitary adenomas, and thick, irregular, and multilocular enhancement is indicative of craniopharyngiomas.

### 2.3. Machine Learning Methods

Logistic Regression: Logistic Regression is a statistical method used for binary classification problems. Despite its name, it is a classification algorithm, not a regression one. Logistic Regression models the probability that a given data point belongs to one of two classes based on one or more input features. It uses a logistic function to make predictions and is widely used for its simplicity and interpretability.

Support Vector Machine (SVM): Support Vector Machine is a supervised machine learning algorithm used for classification and regression tasks. SVM aims to find a hyperplane that best separates data points into distinct classes. It works by maximizing the margin between classes while minimizing classification errors. SVM is effective in handling high-dimensional data and is known for its ability to handle complex datasets.

Random Forest (RF): Random Forest is an ensemble learning algorithm that combines multiple decision trees to make predictions. It is particularly effective for both classification and regression tasks. RF creates a set of decision trees with random subsets of the training data and features. It then combines their predictions to reduce overfitting and improve accuracy [17].

AdaBoost (Adaptive Boosting): AdaBoost is an ensemble learning algorithm that combines weak classifiers to create a strong classifier. It focuses on improving the classification performance of models by giving more weight to incorrectly classified data points in subsequent iterations. AdaBoost iteratively trains weak classifiers and combines their results to create a powerful ensemble model [18].

Despite the existence of numerous machine learning models, there is currently no machine learning model applied to the diagnosis of cystic sellar lesions.

## 3. Materials and Methods

### 3.1. Patients

This study retrospectively included pathologically and intraoperatively confirmed patients with cystic sellar lesions diagnosed at Peking Union Medical College Hospital (PUMCH) from June 2005 to November 2020. The included pathology types were Rathke’s cleft cysts, cystic craniopharyngiomas, cystic pituitary adenomas, and pituitary apoplexy. The criteria included the following: (1) adults with intraoperatively and histopathologically confirmed Rathke’s cleft cysts, cystic craniopharyngiomas, cystic pituitary adenomas, or pituitary apoplexy; (2) preoperative contrast-enhanced T1 (T1-CE)-weighted and T2-weighted MRI; This study design was approved by the Institutional Review Board, and all patients provided informed consent. Finally, 390 patients met the inclusion criteria.

### 3.2. MRI Data Acquisition and Preprocessing

Most preoperative MRI examinations were performed on a 3.0-T MRI scanner (Discovery MR750, Chicago, IL, USA). A minority of patients were examined with a 1.5-T MRI scanner. T1-weighted images (gadolinium chelate, PUMCH standard half dose 0.05 mmol/kg; slice thickness 3–6 mm for 5–20 slices; 10 patients were scanned with 0.5–1.5 mm thickness for 16–75 slices; echo time, 7.264–17 ms; inversion time, 400–613 ms) and T2-weighted images (slice thickness 3.5–6 mm; 7–21 slices; repetition time, 3437–4860 ms; echo time, 79.92–107.648 ms) were obtained. All MR images have a minimum resolution of at least 256 × 256. The original DICOM data were converted to NIfTI format for later processing and anonymity.

### 3.3. Image Segmentation

T2-weighted images were coregistered to T1-CE images for a clear delineation of the tumor boundary as well as the elimination of head movement with ANTs (v2.3.5-30, compiled under Ubuntu 20.04.2 LTS on amd64 architecture) [19]. The registration parameter was the default for rigid bodies. All registered images were checked manually. 

The three-dimensional region of interest (ROI), which included the cysts’ wall and the cysts’ fluid, was automatically segmented initially with in-house software based on U-Net. Cystic pituitary lesions in T1 contrast-enhanced images were manually segmented to build a training set. The model was trained with PyTorch 1.6.0 (https://pytorch.org/) and Python 3.8.8.

All of the automatic segmentations were manually corrected. They were manually delimitated by two neurosurgeons using the ITK-SNAP software (http://www.itksnap.org/pmwiki/pmwiki.php) (accessed on 17 October 2023) [20]. Figure 1 shows the segmentation sample. The ROI was then evaluated by senior neuroradiologists. If the difference between ROIs was ≤5% for the two neurosurgeons, the final ROIs were defined by the overlapping area of the initial ROIs; if the difference between ROIs was >5%, the neuroradiologist made the final decision.

### 3.4. Feature Extraction

The brightness of the T1-CE and T2-weighted images was normalized by centering the voxels at the mean value with standard deviation (SD) based on all gray values using the preset module of PyRadiomics (3.0.1, http://www.radiomics.io/) [21].

We extracted the radiomic features from both 3D images and the slices with maximum lesions with a 2D feature calculation. Considering the thickness of the slices, 3D feature extraction was finished after a resampling of 3mm. Therefore, the Laplace of Gaussian (LoG) filtering should have a sigma greater than 3.0. The resampling size of 2D images was 2mm. The sigma values for 3D images were 3.0 and 5.0, and the values for 2D images were 2.0, 3.0, 4.0, and 5.0.

Wavelet filtering (Coif 1) and LoG filtering were applied. A total of 1037 radiomics features were extracted from the ROIs of three-dimensional images [21]. All radiomics features were scaled based on the SD of the training set to avoid fluctuation.

### 3.5. Feature Selection, Model Construction, and Validation

Radiomics features were selected by the least absolute shrinkage and selection operator (LASSO) in the training dataset with code constructed using Scikit-learn (v0.24.1, http://scikit-learn.org) [22].

The four types of cystic lesions were grouped in pairs. Four types of feature sets were investigated (with or without filtering and 2D or 3D). The machine learning algorithms were support vector machine (SVM), random forest, and AdaBoost (with a decision tree or SVM as its basis).

We used five-fold cross-validation for evaluation. The average area under the receiver operating characteristic curves (AUC) was assessed.

### 3.6. Establishment of Clinical Knowledge-Based Method

Because doctors with different clinical experiences may have variations in their judgments, we have simply designed this model as a multi-stage model to obtain a general diagnosis (Figure 2). First, doctors were required to assess intracystic components. According to the recent study [11], cysts with calcifications are classified as craniopharyngiomas, those with a high fluid level as cystic pituitary adenomas, and those with intracystic nodules as Rathke’s cleft cysts. Then, doctors were required to provide cyst wall enhancement features: absence of enhancement indicates arachnoid cysts; thin and regular enhancement suggests Rathke’s cleft cysts; thick-walled enhancement with septations suggests cystic pituitary adenomas; and thick, irregular, and nodulariform enhancement is indicative of craniopharyngiomas.

### 3.7. Statistical Analysis

Statistical analysis was performed with R 4.0.5 (https://www.r-project.org) and Python 3.8.8 (https://www.python.org).

## 4. Results

### 4.1. Patient Characteristics and Pathology Types

The enrolled patients were mainly female, with an age of 41.28 years (±15.08). The proportion of men with cystic craniopharyngioma was slightly higher (54.1%), and the proportion of women with other diseases was higher. In terms of age, the medians and averages of the groups were close (average 41, median 40). Patients with cystic craniopharyngioma tend to have a high proportion of young patients, but none of the other types (see Table 1 and Table 2 for baseline levels and Figure 3 for the histogram of age distribution).

### 4.2. Image Segmentation and Feature Selection

This research carried out a total of 4 types of feature extraction methods: Three-dimensional features containing wavelet filtering and edge detection filtering, 2D features containing wavelet filtering and edge detection filtering, 3D features containing only basic features, and 2D features containing only basic features. The first two groups are common processing strategies in radiomics, and the latter two groups aim to make the included features more intuitive and improve their interpretability. The extracted features are summarized in Table 3. The heatmap of radiomics features is shown in Figure 4. After feature selection, the most significant differential features were ranked, and the top three influential omics features were listed in Table 4. It can be observed that the top three ranked features are predominantly T2 features, with no inclusion of T1 features. This indicates that T2 imaging exhibits high diagnostic performance in the diagnosis, aligning with our diagnostic focus on T2 imaging for sellar cystic lesions.

### 4.3. Radiomics Model Validation and Model Comparison

The performance of the machine learning algorithms is shown in Table 5. For the Mann–Whitney U test for each feature separately, there were hundreds of features satisfying *p* < 0.05 in each group. Therefore, in this study, the Mann–Whitney U test alone did not reduce the number of features.

To evaluate the ability of a single feature to distinguish different lesions, we used a single feature as a single index classifier for classification. It was worth noting that whether in the 2D or 3D feature extraction mode, the distinguishment of Rathke’s cleft cyst and cystic craniopharyngioma or Rathke’s cleft cyst and pituitary apoplexy had many single features with an AUC value > 0.80, while there was no single feature with an AUC value > 0.70 in the distinguishment of pituitary apoplexy and cystic pituitary adenoma.

### 4.4. Comparison with Clinical Knowledge Base Methods

Although some models achieved an AUC value of 0.8, to better illustrate the clinical applicability of the model, we compared the model with the highest diagnostic capability against experienced clinicians. To mitigate potential bias introduced by clinical expertise, we employed a decision tree approach [16], where clinical doctors assessed the presence of intracystic components and cyst wall enhancement features. The final diagnosis was based on the decision tree. The results showed that, across all six tasks, machine learning models outperformed human judgments, with an average accuracy rate approximately 8% higher than that of doctors (Table 6).

## 5. Discussion

Among the various basic statistical machine learning algorithms, AUC values > 0.70 can be achieved in the identification of most lesions. Support vector machine-based models performed the best in each comparison group, whether they were used directly or integrated with AdaBoost or Bagging methods. Logistic regression was performed secondary to the support vector machine. It suggested that classical machine learning algorithms still performed stably and reliably. Decision tree-based algorithms performed relatively poorly. They are accompanied by the disadvantage of overfitting [23]. This was responsible for their relatively poor performance in this study.

The preoperative diagnosis of cystic lesions in the sellar region is crucial for surgical planning. For arachnoid cysts and Rathke’s cleft cysts, sometimes only fenestration and partial resection are necessary. However, for pituitary adenomas and craniopharyngiomas, total resection is the primary surgical approach, aiming to achieve a high rate of complete removal to reduce tumor recurrence. Craniopharyngiomas, in comparison to pituitary adenomas, have a less fixed location but possess a distinct capsule structure that allows for complete excision along the capsule. The preoperative diagnostic accuracy in clinical practice is only 0.67, indicating ongoing challenges in the differential diagnosis of such conditions. This study compared clinical knowledge-based methods and found that the proposed machine learning model based on radiomics outperformed clinical doctors’ diagnostic criteria across all tasks. This suggests that the model can assist in clinical diagnosis to a certain extent and aid in formulating surgical plans.

In traditional machine learning models of each group, Rathke’s cleft cyst and cystic craniopharyngioma, cystic pituitary adenoma, and craniopharyngioma were relatively easy to identify by all algorithms. Rathke cleft cyst and cystic pituitary adenoma, pituitary apoplexy, and Rathke cleft cyst, pituitary apoplexy, and craniopharyngioma performed moderately. It is hard to distinguish pituitary apoplexy from cystic pituitary adenoma in any model. The average accuracy values of different models for various lesion combinations evaluated in leave-one-out validation (LOOV) were consistent with the model performance and effect ranking of five-fold cross-validation.

Among the comparison groups, it is very difficult to differentiate cystic pituitary adenoma from pituitary apoplexy, and the performances of all models were poor. Necrosis and cystic degeneration occur in 5–18% of pituitary tumors, a process often associated with apoplexy [24,25]. In studies of cystic fluid components in cystic pituitary adenomas, most cystic fluids have detectable levels of hemoglobin, and some cystic fluids have a high level of hormones [26]. Secretory cystic degeneration and hemorrhagic cystic degeneration may coexist in cystic pituitary adenomas. In a study including 14 patients initially evaluated as having cystic pituitary adenomas, after careful verification, only 2 were finally confirmed to have cystic pituitary adenomas without hemorrhagic changes, and the remaining patients’ lesions had hemorrhagic components [27]. This finding suggests that a large proportion of cystic pituitary adenomas may have hemorrhagic manifestations, which is also consistent with the diagnosis of pituitary apoplexy in pathology. Based on these previous studies and the difficulties in image identification in this study, we hypothesize that some cystic pituitary adenomas form gradually after apoplexy, and the other components may arise from the secretion of the cyst wall. Therefore, cystic pituitary adenomas and pituitary apoplexy were hard to distinguish from MRI.

Cystic craniopharyngioma and Rathke’s cleft cyst, cystic craniopharyngioma, and cystic pituitary adenoma performed well in most predictive models. These results suggest that there are clear differences in their imaging. Previous studies have described the image identification of craniopharyngioma in terms of volume, calcification, and shape [28]. That is the basis for its outstanding differences in radiomic features. Generally, it is believed that the volume of the craniopharyngioma is important for identification. Although a considerable number of craniopharyngiomas in this study are larger in size than Rathke’s cleft cysts, their main volume distribution ranges are similar.

In this study, the traditional machine learning models construct four feature groups according to the dimension, with/without LoG processing or wavelet filtering. The performance of each group of models for discrimination is roughly similar, and the trend of the difficulty of discriminating different combinations of lesions is the same. This indicates that although the 3D features cover more layers and more information, the information provided by the 2D maximum layer is sufficient for radiomic identification.

Possible future research directions in this field include a more detailed description of lesions and the differentiation of pathological subtypes. If more molecular markers could be found in the future, their imaging manifestations are also expected to be further investigated. Recently, there have been many advances in digital image processing methods, such as attention mechanisms and capsule networks [29,30]. Because of the lagging of their applications in clinical medicine, these methods have not been fully utilized. If they can be properly used in image analysis of cystic lesions of the sellar region, it will hopefully unearth more information and make more accurate predictions.

## 6. Limitations

There are several limitations to this study. First, this study is a retrospective study rather than a prospective one. This could introduce various biases, including selection bias and confounding bias. For example, all included patients were required to undergo MRI scans at our hospital, while those with MRI scans conducted at external institutions were excluded from this study. As a result, the included population consisted primarily of newly diagnosed patients. Second, this study focused only on preoperative imaging features, without prognosis. Third, some rare but non-negligible pathological types were not covered in this study. Fourth, patients were collected from a single center, and an external dataset should be collected for future study and evaluation.

## 7. Conclusions

In this study, the image differential diagnosis of cystic lesions of the sellar region was performed, and radiomics methods were used with T1CE and T2WI sequences. Results showed that the model achieved an average AUC value of 0.7685 and an average Accuracy of 0.7532, which outperformed the traditional clinical knowledge-based method by about 8%. In conclusion, we first provide a non-invasive method based on radiomics and machine learning methods for accurately differentiating between these lesions, which could improve preoperative diagnosis accuracy and help to make surgery plans in clinical work.

## Figures and Tables

**Figure 1 bioengineering-10-01295-f001:**
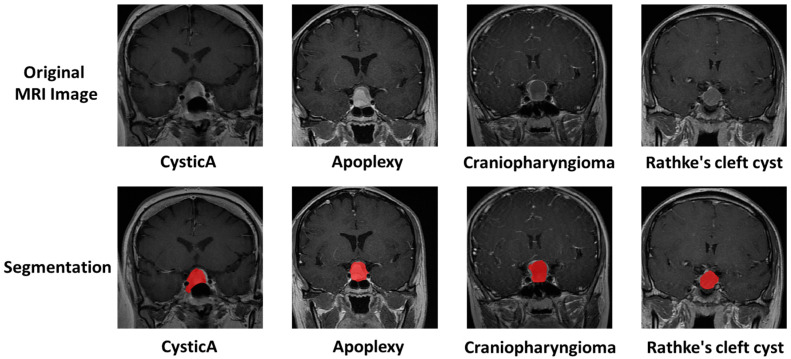
Sample of the segmentation after automatic labeling and manual quality control. Four typical cystic sellar lesions on T1CE were shown, including cystic pituitary adenoma (cysticA), pituitary apoplexy, cystic craniopharyngioma, and Rathke’s cleft cyst. The red mark was superimposed on the segmented area.

**Figure 2 bioengineering-10-01295-f002:**
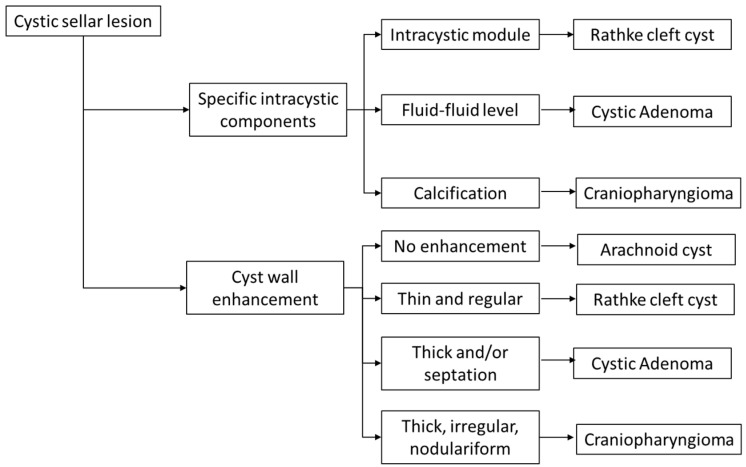
Clinical knowledge-based model to differentiate cystic sellar lesions [11].

**Figure 3 bioengineering-10-01295-f003:**
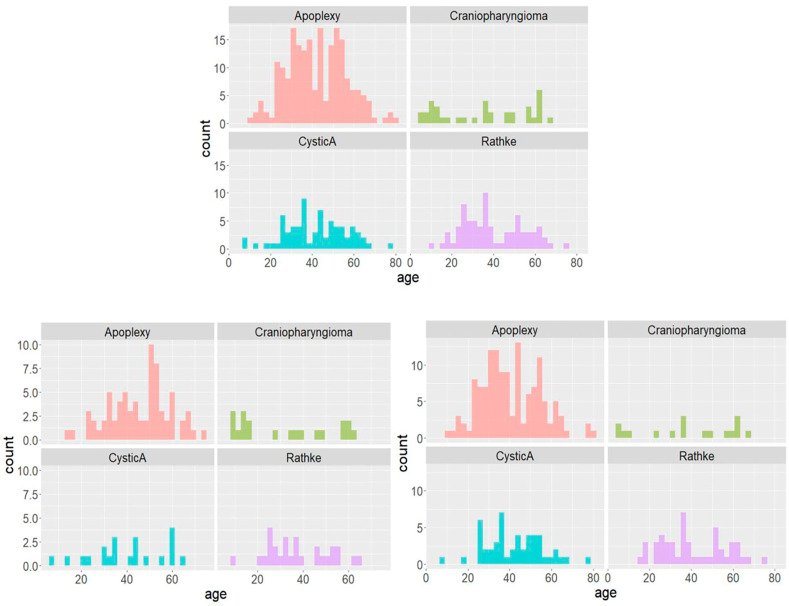
Histogram of age distribution, top: all patients, bottom left: male patients, bottom right: female patients.

**Figure 4 bioengineering-10-01295-f004:**
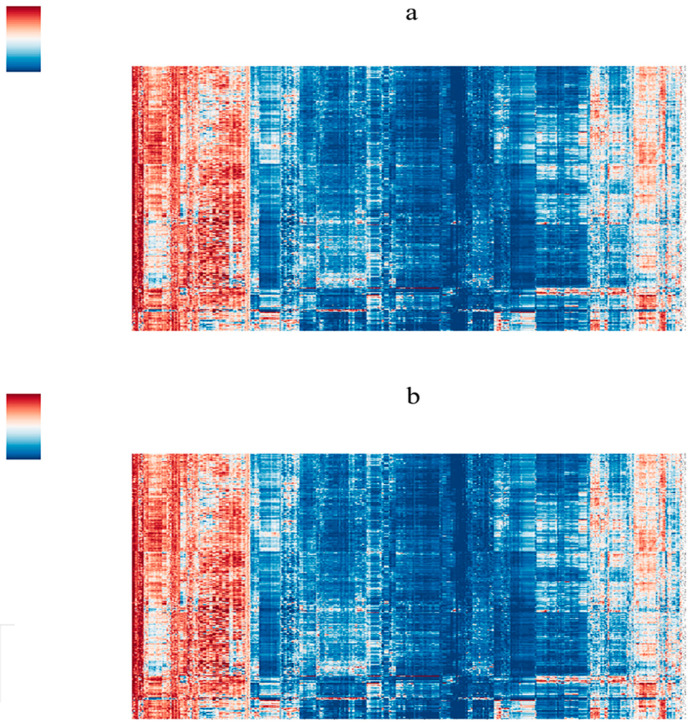
Heatmap: the horizontal axis was the radiomics feature, and the vertical axis was the case. (**a**) 3D features; (**b**) 2D features.

**Table 1 bioengineering-10-01295-t001:** Baseline of the patients.

Type	*n*	Age Mean (SD)	Female (%)	Male (%)
Sum	390	41.06 (15.14)	249 (63.8)	141 (36.2)
Apoplexy	205	42.38 (14.05)	133 (64.9)	72 (35.1)
Craniopharyngioma	37	35.30 (21.00)	17 (45.9)	20 (54.1)
Cystic Adenoma	71	41.58 (14.54)	50 (70.4)	21 (29.6)
Rathke’s Cleft Cyst	77	39.82 (14.72)	49 (63.6)	28 (36.4)
*p*-value			0.094	

**Table 2 bioengineering-10-01295-t002:** Age Distribution of the patients.

Age	*n*	Mean	SD	Median	p25	p75	Min	Max	Skew	Kurt
Sum	390	41	15	40	30	53	5	80	−0.03	−0.56
Apoplexy	205	42	14	42	32	53	10	80	0.13	−0.48
Craniopharyngioma	37	35	21	37	14	57	5	66	−0.001	−1.6
Cystic Adenoma	71	42	15	42	32	52	7	77	−0.077	−0.24
Rathke’s Cleft Cyst	77	40	15	36	28	53	10	75	0.28	−0.85

**Table 3 bioengineering-10-01295-t003:** Summary of Feature Extraction.

		3D		2D	
		T1CE	T2WI	T1CE	T2WI
Shape		14	14	9	9
First-order		18	18	18	18
Texture	GLCM	24	24	24	24
	GLRLM	16	16	16	16
	GLSZM	16	16	16	16
	GLDM	14	14	14	14
	NGTDM	5	5	5	5
LoG		186	186	372	372
Wavelet		744	744	186	186
Total		1037	1037	660	660

**Table 4 bioengineering-10-01295-t004:** Top three ranked features.

Apoplexy	Apoplexy	Apoplexy	CysticA	Rathke	Rathke
Craniopharyngioma	CysticA	Rathke	Craniopharyngioma	Craniopharyngioma	CysticA
T1CE_wavelet-LHH_ngtdm_Complexity	T1CE_original_shape_Elongation	T2RS_original_glcm_MCC	T1CE_wavelet-LLL_ngtdm_Complexity	T2RS_log-sigma-5-0-mm-3D_glcm_Idn	T1CE_wavelet-LLH_ngtdm_Coarseness
T1CE_wavelet-LLL_glcm_ClusterProminence	T1CE_original_shape_Flatness	T2RS_log-sigma-3-0-mm-3D_glcm_Correlation	T2RS_original_gldm_LargeDependenceHighGrayLevelEmphasis	T2RS_wavelet-LHL_glcm_MCC	T1CE_wavelet-LLL_glcm_Correlation
T2RS_original_gldm_LargeDependenceHighGrayLevelEmphasis	T1CE_original_shape_LeastAxisLength	T2RS_log-sigma-3-0-mm-3D_glcm_JointAverage	T2RS_log-sigma-5-0-mm-3D_firstorder_Skewness	T2RS_wavelet-HLH_glcm_MCC	T2RS_original_firstorder_Skewness

**Table 5 bioengineering-10-01295-t005:** Model performances. AUCs were compared between different methods.

Compare	Method	3D	3D w/o Filters	2D	2D w/o Filters
Apoplexy vs. Craniopharyngioma	Random Forest	0.6591	0.6682	0.6306	0.5986
	Bagging SVM	0.6858	0.7019	0.5855	0.5895
	SVM	0.6871	0.7011	0.6312	0.6228
	AdaBoost DecisionTree	0.6140	0.5666	0.5628	0.5974
	AdaBoost	**0.7001**	0.7070	0.6460	0.6110
	Logistic Regression	0.6890	**0.7102**	**0.6660**	**0.6435**
Apoplexy vs. CysticA	RandomForest	**0.6454**	0.5556	0.6104	0.6300
	BaggingClassifier_SVM	0.6108	0.5367	0.6320	0.6465
	SVM	0.6249	0.5719	0.6520	0.6524
	AdaBoost_DecisionTree	0.5911	0.5801	0.5914	0.5044
	AdaBoost	0.6113	**0.6141**	**0.6641**	0.6593
	Logistic_Regression	0.6060	0.5796	0.6406	**0.6624**
Apoplexy vs. Rathke	RandomForest	0.7820	0.7944	0.7787	**0.7932**
	BaggingClassifier_SVM	**0.8046**	0.7721	0.7775	0.7385
	SVM	0.8025	**0.8032**	0.7697	0.7728
	AdaBoost_DecisionTree	0.6647	0.6043	0.6451	0.6276
	AdaBoost	0.7902	0.7903	**0.7827**	0.7807
	Logistic_Regression	0.7899	0.7883	0.7712	0.7408
CysticA vs. Craniopharyngioma	RandomForest	0.7387	**0.7695**	0.7161	0.7209
	BaggingClassifier_SVM	0.7989	0.7308	0.6982	**0.7769**
	SVM	0.7607	0.7610	**0.7770**	0.7737
	AdaBoost_DecisionTree	0.6544	0.6377	0.7274	0.5924
	AdaBoost	**0.8096**	0.7452	0.7468	0.7438
	Logistic_Regression	0.7598	0.7202	0.7376	0.7237
Rathke vs. Craniopharyngioma	RandomForest	**0.8263**	**0.8355**	0.8348	0.7842
	BaggingClassifier_SVM	0.8141	0.8217	0.8451	0.8122
	SVM	0.8176	0.8165	**0.8534**	0.8509
	AdaBoost_DecisionTree	0.7506	0.6155	0.7358	0.7871
	AdaBoost	0.8224	0.8213	0.8534	0.8522
	Logistic_Regression	0.8085	0.8165	0.8534	**0.8584**
Rathke vs. CysticA	RandomForest	0.6701	0.6716	0.6917	0.6949
	BaggingClassifier_SVM	**0.7660**	0.6859	0.7019	0.6440
	SVM	0.7511	**0.6939**	0.7028	0.6570
	AdaBoost_DecisionTree	0.6418	0.6053	0.6149	0.5832
	AdaBoost	0.7506	0.6798	**0.7361**	**0.7006**
	Logistic_Regression	0.7235	0.6759	0.7254	0.6635

**Table 6 bioengineering-10-01295-t006:** Comparison between proposed methods and clinical knowledge-based methods.

	Machine Learning	Clinical Knowledge-Based Method
Apoplexy vs. Craniopharyngioma	0.7708	0.6876
Apoplexy vs. CysticA	0.686	0.5404
Apoplexy vs. Rathke	0.7633	0.7493
CysticA vs. Craniopharyngioma	0.7675	0.5823
Rathke vs. Craniopharyngioma	0.8293	0.8135
Rathke vs. CysticA	0.7022	0.6758
Mean Accuracy	0.7532	0.6748

## Data Availability

The data and code used to support the findings of this study are available from the corresponding author on request.

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
