# Peer review of "Machine Learning Approaches to Differentiate Sellar-Suprasellar Cystic Lesions on Magnetic Resonance Imaging"

_bioengineering, 2023, doi:10.3390/bioengineering10111295_

Round 1

Reviewer 1 Report

It is not clear what types of preprocessing steps have been taken to normalize the images.

What was the input format? Was it DICOM?

Has any type of alignment or registration been performed before segmentation?

Has resizing been done before segmentation?

How many reviewers were involved in the correction of the segmentation? If there were multiple reviewers, what was the interrater reliability of the reviewers?

What was your approach to resolving the class imbalance? Did you use oversampling or under-sampling?

Radiomics features can often be correlated or redundant, which may not add substantial information to the model. Identifying and removing redundant features is a non-trivial task. How did you manage redundancy?

Radiomics models developed on one dataset may not generalize well to different populations or clinical settings, necessitating external validation. Have you tested the model on a holdout set?

In computer vision, especially in image analysis tasks, Lasso operates on individual features (e.g., pixel intensities), and it may not capture spatial dependencies or patterns in the data. What was your solution to minimize this limitation?

Lasso tends to produce sparse feature sets, which means it selects only a subset of the available features and sets the coefficients of many other features to zero. In some cases, it may select too few features, leading to underfitting. How did you mitigate the negative impact of Lasso as a feature selection tool in this study?

Was the AUC value of 0.7685 the test set performance? If so, what was the cross-validated performance?

What were the PR AUC values for both the test and train sets?

AUC PR and AUC ROC figures should be provided

Reviewer 2 Report

The manuscript sounds technically average; however, I have following concerns should be addressed before any decision.  

1.      Please explain in your captions of figure and title of table, why are these tables or figures necessary in your paper? What are the purposes and what are the message you want to deliver via these figures and tables?

2.      The current metrics might not be sufficient to judge the performance of the model holistically. Please enhance the result analysis part of your paper.

3.      The existing literature should be classified and systematically reviewed, instead of being independently introduced one-by-one.

4.       In the introduction section, the motivations of the proposed access control model must be included in detail. The section numbering must be changed in the paper organization paragraph.

5.      The abstract is too general and not prepared objectively. It should briefly highlight the paper's novelty as what is the main problem, how has it been resolved and where the novelty lies?

6.      The 'conclusions' are a key component of the paper. It should complement the 'abstract' and normally used by experts to value the paper's engineering content. In general, it should sum up the most important outcomes of the paper. It should simply provide critical facts and figures achieved in this paper for supporting the claims.

7.      For better readability, the authors may expand the abbreviations at every first occurrence.

8.      The author should provide only relevant information related to this paper and reserve more space for the proposed framework.

9.      The theoretical perceptive of all the models used for comparison must be included in the literature.

10.   What are the real-life use cases of the proposed model? The authors can add a theoretical discussion on the real-life usage of the proposed model.

11.   The related works section is very short and no benefits from it. I suggest increasing the number of studies and add a new discussion there to show the advantage.  

12.   The descriptions given in this proposed scheme are not sufficient that this manuscript only adopted a variety of existing methods to complete the experiment where there are no strong hypothesis and methodical theoretical arguments. Therefore, the reviewer considers that this paper needs more works.

13.   Key contribution and novelty has not been detailed in manuscript. Please include it in the introduction section

Minor corrections needed.

Reviewer 3 Report

The study aimed to differentiate various pathological types of cystic lesions in the sellar region using preoperative magnetic resonance imaging (MRI). The study used radiomics, machine learning, and artificial neural network approaches with contrast-enhanced MRI to distinguish between different types of cystic lesions.

 The main limitations of the study are as follows:

1.     The model could not differentiate between pituitary apoplexy and cystic pituitary adenoma (cysticA) using the applied methods. The area under the curve (AUC) value for distinguishing these two types of lesions was only 0.6641, indicating that the classification was almost unachievable with the algorithms and feature sets used in the study.

2.     The study was retrospective, which may introduce potential biases and limit the generalizability of the findings.

3.     The study's sample size was limited to 399 patients, which may not be sufficient to fully capture the variability and complexity of cystic lesions in the sellar region.

4.     The study focused on four specific subtypes of cystic lesions (pituitary apoplexy, cystic pituitary adenoma, Rathke's cleft cyst, and cystic craniopharyngioma), which may not cover all possible types of cystic lesions in the sellar region.

5.     The study relied on preoperative MRI data, which may not provide a complete picture of the lesions' characteristics and may be subject to variability in image quality and acquisition parameters.

6.     The study used a single-center dataset, which may limit the generalizability of the results to other clinical settings and populations.

7.     The study did not explore the potential impact of different MRI scanners and imaging protocols on the performance of the machine learning models. Variability in image acquisition and quality across different scanners and institutions could affect the reproducibility and robustness of the models.

8.     The study did not validate the models on an independent external dataset, which is crucial for assessing the true performance and generalizability of the models in real-world clinical practice.

9.     The study did not investigate the potential impact of other clinical factors, such as patient demographics, medical history, and laboratory results, on the differentiation of cystic lesions. Incorporating such information could potentially improve the performance of the models.

good
